# Family-based interventions to increase physical activity in children: a meta-analysis and realist synthesis protocol

Helen Elizabeth Brown,[1,2] Andrew J Atkin,[2] Jenna Panter,[1,2] Kirsten Corder,[1,2] Geoff Wong,[3] Mai J M Chinapaw,[4] Esther van Sluijs[1,2]

[1]Department of MRC Epidemiology, University of Cambridge School of Clinical Medicine, Institute of Metabolic Science, Cambridge, UK
[2]UKCRC Centre for Diet and Activity Research (CEDAR), Institute of Public Health, University of Cambridge, Cambridge, UK
[3]Centre for Primary Care and Public Health, University of London, London, UK
[4]Department of Public and Occupational Health, EMGO Institute for Health and Care Research, VU University Medical Centre, Amsterdam, The Netherlands

**Correspondence to**
Dr Helen Elizabeth Brown; heb56@medschl.cam.ac.uk

## ABSTRACT

**Introduction:** Despite the established relationship between physical activity and health, data suggest that many children are insufficiently active, and that levels decline into adolescence. Engaging the family in interventions may increase and maintain children's physical activity levels at the critical juncture before secondary school. Synthesis of existing evidence will inform future studies, but the heterogeneity in target populations recruited, behaviour change techniques and intervention strategies employed, and measurement conducted, may require a multifaceted review method. The primary objective of this work will therefore be to synthesis evidence from intervention studies that explicitly engage the family unit to increase children's physical activity using an innovative dual meta-analysis and realist approach.

**Methods and analysis:** Peer-reviewed studies will be independently screened by two authors for inclusion based on (1) including 'healthy' participants aged 5–12 years; (2) having a substantive intervention aim of increasing physical activity, by engaging the family and (3) reporting on physical activity. Duplicate data extraction and quality assessment will be conducted using a specially designed proforma and the Effective Public Health Practice Project Quality Assessment Tool respectively. STATA software will be used to compute effect sizes for meta-analyses, with subgroup analyses conducted to identify moderating characteristics. Realist syntheses will be conducted according to RAMESES quality and publication guidelines, including development of a programme theory and evidence mapping.

**Dissemination:** This review will be the first to use the framework of a traditional review to conduct a dual meta-analysis and realist synthesis, examining interventions that engage the family to increase physical activity in children. The results will be disseminated through peer-reviewed publications, conferences, formal presentations to policy makers and practitioners and informal meetings. Evidence generated from this synthesis will also be used to inform the development of theory-driven, evidence-based interventions aimed at engaging the family to increase physical activity levels in children.

**Protocol registration:** International Prospective Register for Systematic Reviews (PROSPERO): number CRD42013005780.

## BACKGROUND

The relationship between physical activity and health is well established; in children, physical activity is associated with cardiovascular risk factors,[1 2] body composition (particularly waist circumference and fat mass)[3 4] and bone health.[5] Engaging regularly in physical activity has a beneficial effect on depression, anxiety and self-esteem, and may also be associated with improved cognitive performance and scholastic achievement in this age group.[6] Despite this, data from several countries suggest that the majority of children are insufficiently active to confer health benefit[7 8]; and that levels of physical activity decline substantially throughout childhood and into adolescence.[9 10]

Intervention prior to this age-related decline (commonly between 9 and 11 years of age) may be important to maintain adequate physical activity levels. Observational evidence suggests that a steeper decline in physical activity occurs out of school time, particularly at the weekend,[11] and so intervention during these periods may represent the most responsible use of limited public health resources.[12] However, the school environment is the most frequently used setting for researchers trying to improve children's physical activity patterns; as such, studies have been largely unsuccessful.[13 14]

A recent National Institute for Health and Care Excellence review identified characteristics of successful interventions, highlighting

those based in the home as most effective in changing health behaviours.[15] [16] Parental support has been consistently and positively associated with increased physical activity in children.[17] The addition of a family component (eg, parent education) to school-based interventions has also proved to be efficacious.[14] Engaging the family may therefore be a promising strategy, and may support the maintenance of physical activity levels (and concurrently avoid an increase in sedentary behaviour) at the critical juncture before children progress to secondary school.[14] [18]

Currently, only one systematic overview with a focus on family-based interventions is available. O'Connor et al[19] reviewed literature published before January 2008, concluding that the large number of pilot studies, and the variability in study design and outcome measures, restricted insight into how best to engage parents in physical activity promotion. Other broader reviews have included subsections on family-based interventions, reaching similar conclusions.[18] [20] To develop effective interventions, a more thorough understanding of such studies is required. A brief scoping review identified a number of interventions published within the last 5 years, indicating the need for an update in this field. This may include mediation analysis to explore causal mechanisms, and process evaluation to examine the implementation, receipt and setting of an intervention and support interpretation of results.[21]

A synthesis of existing evidence would inform future family-based studies aimed at improving and maintaining physical activity levels in children. However, the heterogeneity in target populations recruited, behaviour change techniques and intervention strategies employed, measurement conducted and number of pilot studies included, may require a multifaceted approach. Well-conducted meta-analyses allow for an objective appraisal of the evidence, provide a precise estimate of a treatment effect, and may explain heterogeneity between the results of individual studies.[22] [23] Meta-analytic techniques are, however, limited to quantitative analysis, which may not be sufficient to understand *how* an intervention operates. In contrast, a realist synthesis offers no quantifiable summary of intervention effectiveness, but considers the interaction between context, mechanism and outcome, exploring 'what works for whom, under what circumstances, how and why?'[24] Previous reviews in this area have been somewhat constrained by these limitations, and therefore provide insufficient insight into the complex causal pathways that may underpin interventions.

Combining these two methodologies, using the framework of a traditional systematic review, may provide a more comprehensive examination of studies. The primary objective of this work will therefore be to summarise, using the innovative dual meta-analysis and realist synthesis approach, existing peer-reviewed intervention studies that explicitly engage the family unit to increase physical activity in children.

## METHODS

The protocol is registered with the International Prospective Register for Systematic Reviews (PROSPERO) CRD42013005780.

Given the multifaceted approach planned, the methods for this work will be divided by process. The first section will detail overall review methods that relate to all studies, including search strategy, study screening, inclusion and exclusion criteria, quality assessment and general data extraction. The second section, headed *Realist Synthesis*, will outline data extraction and analysis procedures specific to the realist synthesis. The third section, headed *Meta-analysis*, will provide complementary information related to the meta-analysis.

### Search strategy

The following databases will be searched for articles published up to and including June 2013, with no limit on earliest year of publication: PubMed (title and abstract), Web of Knowledge (topic), Scopus (title, abstract and keywords), Ovid MEDLINE (abstract), PsycInfo (abstract). The search strategy will be common across databases, and will consist of PICO terms; *participants* of interest, *intervention*, *control* and primary *outcome* of interest. Full details of search terms used will be reported using figure 1.

In addition to database searches, reference lists of included full-text articles and personal archives will be screened for studies meeting inclusion criteria. Four review articles, focusing on interventions in a range of settings, have been identified and will also be used to source additional studies.[14] [18–20]

### Study screening

All retrieved titles and abstracts will be screened by the primary author (HEB), and duplicate screened by another author (AJA or EvS). Criteria for screening will be refined if necessary, and any discrepancy in inclusion or exclusion will be resolved through a consensus discussion among the three authors (HEB, EvS and AJA). Full text versions of selected articles will then be obtained, and inclusion and exclusion criteria assessed (following the same procedure as for titles and abstracts; duplicate screening and consensus discussion between HEB, EvS and AJA).

### Inclusion and exclusion criteria

Studies will be included that report on interventions that (1) have an explicit aim to increase physical activity, (2) use the family as the setting, (3) include healthy children aged 5–13 years at baseline, (4) report on results from a valid measure of physical activity and (5) were published in a peer-reviewed journal up to and including June 2013. Study designs may include randomised controlled trials, comparison trials and/or quasiexperimental studies. Irrespective of the over-arching aim (eg, to reduce weight), if a substantive part of the intervention targets the determinants of physical activity, or aims to

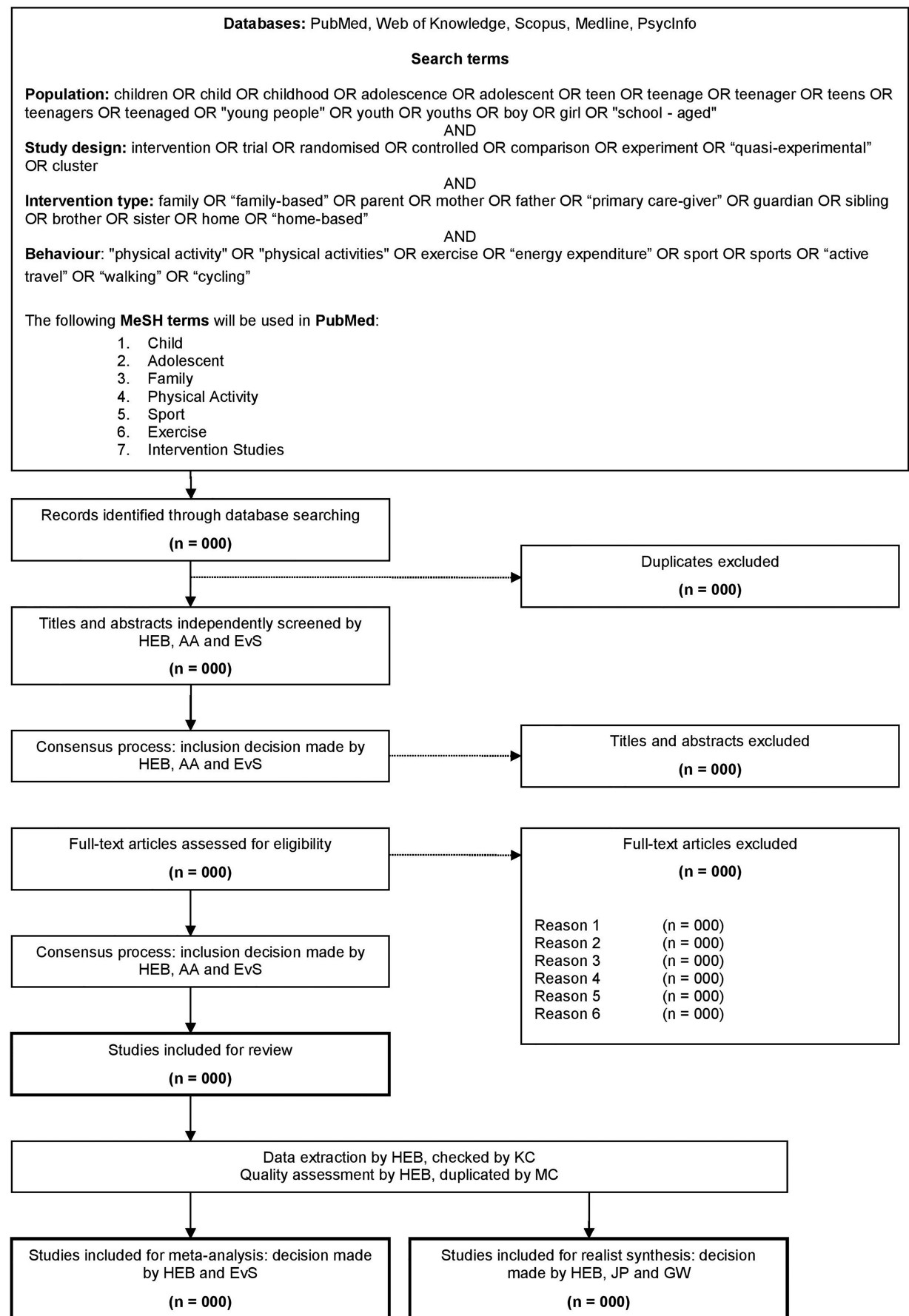

**Figure 1** Search strategy for review of existing intervention 1 studies.

**Table 1** Results of duplicate quality assessment of studies, using the EPHPP Quality Assessment Tool for Quantitative Studies

| Study authors and date | Intervention name (where available) | Selection bias | Study design | Confounders | Blinding | Data collection methods | Withdrawal and drop-out | Quality assessment rating | Overall score/12 |
|---|---|---|---|---|---|---|---|---|---|
| Total studies scoring 'strong' per criterion | | n | n | n | n | n | n | n | n |
| 000 | 000 | Strong/moderate/weak | Strong/moderate/weak | Strong/moderate/weak | Strong/moderate/weak | Strong/moderate/weak | Strong/moderate/weak | Strong/moderate/weak | n |
| 111 | 111 | Strong/moderate/weak | Strong/moderate/weak | Strong/moderate/weak | Strong/moderate/weak | Strong/moderate/weak | Strong/moderate/weak | Strong/moderate/weak | n |

EPHPP, Effective Public Health Practice Project.

modify physical activity in any way, it will be included. Studies that engage the family as one of a range of settings, and that isolate the effect by evaluating the family-only intervention separately, will be included. Studies that do not have a substantive role for families in the physical activity component (ie, parent newsletters only) will be excluded. Studies that recruited samples who were overweight or from the general population (ie, whole school or community) will be included, but those in which participants were defined as 'obese' (by whichever definition the authors put forward) will be excluded. Valid measures of physical activity (either as total activity, or during specific periods of the day or week) may include those measured in free living by self-report or proxy-report questionnaire or diary, pedometer, accelerometer, inclinometer or heart rate monitor. These exclusion criteria are definitive, to ensure search results are reflective of specific review objectives. Results of the duplicate screening will be reported using figure 1.

Those studies eligible for general review may also be included in the meta-analysis, the realist synthesis, or both. This will result in three separate but overlapping groups of interventions: (1) studies for general review, (2) studies for realist synthesis and (3) studies for meta-analysis. Studies that report detailed intervention descriptions, or that explore intervention mechanisms (either in analysis or discussion), will be included in the realist synthesis. Studies reporting intervention effect on physical activity (with mean, SD and number of participants analysed), in both intervention and control groups, will be included in the meta-analysis. Those studies that do not meet specific inclusion criteria for either realist synthesis or meta-analysis will be included in a narrative summary of the literature.

### Quality assessment

Quality assessment will be conducted by the lead author using the Effective Public Health Practice Project Quality Assessment Tool for Quantitative Studies; this will be duplicated by an additional author (MJMC) and inter-rater reliability reported as a percentage of items without initial consensus. The Effective Public Health Practice Project tool rates studies as 'strong', 'moderate' or 'weak', using six scales (selection bias, study design, confounders, blinding, data collection methods and withdrawals and drop-outs). Studies are then rated to give an aggregate overall score of 'strong', 'moderate' or 'weak' ('strong' if no 'weak' individual-scale ratings are designated, 'moderate' if one, and 'weak' if two or more). The tool has been recommended for use in assessing public health interventions based on acceptable content and construct validity.[25] Quality assessment results will be reported using table 1.

### Data extraction

Key data regarding study participants, intervention setting and characteristics, and outcomes will be extracted by the primary author (HEB), and checked

for accuracy by another author (KC). Discrepancies will be resolved through consensus discussion. Descriptive data on all studies will be reported using table 2 (table headings detail the information to be extracted).

## REALIST SYNTHESIS

Procedures for the realist synthesis will be informed by the output of the RAMESES Project, which collated expert input and evidence review into comprehensive guidance and publication standards.[24] The primary function of the realist synthesis will be to attempt to explain the outcome patterns observed in included studies. This analysis will address the key realist questions; "what works for whom, under what circumstances, how, and why?"

An initial programme theory will first be developed through a consensus process facilitated by GW, utilising the content expertise of all authors. This programme theory will describe the context and mechanisms necessary to elicit a specified outcome (namely increased physical activity in children). The inclusion of data to inform programme theory development will be guided by the principles of 'relevance' and 'rigour' (as identified by RAMESES output standards,[24 26]) will then be extracted and coded using nVivo qualitative analysis software (first by one author (HEB); and then checked by a second author (JP)). For each 'stage' (ie, text box, or connecting arrow) of the theory, inferences will be made about the possible realist explanation (ie, the context within which change may be triggered, or the mechanisms which precede change). We will also consider how context, mechanism and outcome may interact for specific intervention strategies (eg, goal-setting). Where necessary, supplementary data will be accessed (eg, adding process evaluation or protocol papers of included studies).

Data extracted will be mapped against the initial programme theory to identify areas of strength, and areas that require further research. The programme theory will then be refined to reflect mechanisms that are supported by evidence (eg, arrow thickness may be used to reflect the relative strength of the evidence, and dashed connecting lines may indicate hypothesised configurations of context, mechanism and outcome). If appropriate, existing substantive theory to corroborate stages of the programme theory will be sought.

## META-ANALYSIS

Additional data (specific information for total physical activity, intensity-specific physical activity or domain-specific physical activity) will be extracted for meta-analysis, checked for accuracy by a second author (KC). The type of data extracted (eg, relevant model statistics, coefficients, between-group mean differences, within-group means) will differ according to the data reported in each paper. We will also collect the corresponding measures of precision (eg, SDs, SEs, 95% CIs) and number of participants analysed. Each study will contribute one overall effect size, and we will extract information on potential moderators (eg, studies stratified by sex, ethnicity, weight status) for later subgroup analyses where possible.

To account for the differing unit of measurement we anticipate across studies, we will calculate standardised effect sizes. Intervention effects for each study will likely be represented by the standardised mean difference in outcome, calculated as the physical activity level at follow-up adjusted for physical activity level at baseline (or mean activity change from baseline) divided by the pooled SD of change in physical activity from baseline. Effect sizes for all included studies will then be combined using a random-effects model to derive an overall summary (average) effect estimate (and 95% CI) and assessed against previously used criteria (ie, overall standardised mean difference of −0.2 is considered small, −0.5 moderate and −0.8 large.[27]). Predefined subgroup analyses will be conducted (age, sex, ethnicity and weight status), and issues of outlier and publication bias considered. All computation and analysis of data will be conducted using STATA software by one author (HEB), in collaboration with a second author (EvS). Results of the meta-analysis will be reported using a forest plot of the overall standardised mean difference.

**Table 2** Descriptive characteristics of included studies

| Study (first author; year of publication; country) | Design (study design; level of randomisation where applicable) | Participants (n, children analysed; mean years of age±SD at baseline; % male; mean BMI±SD at baseline) | Physical activity measurement (assessment period; measure used; outcomes reported) | Intervention (intervention name; description) | Intervention (intervention duration; delivery; theoretical grounding) | Control (where applicable, description of any procedure) | Outcome (PA change and statistical tests reported) |
|---|---|---|---|---|---|---|---|
| 000 | 000 | 000 | 000 | 000 | 000 | 000 | 000 |
| 111 | 111 | 111 | 111 | 111 | 111 | 111 | 111 |

BMI, body mass index; PA, physical activity.

## DISCUSSION

This review will be the first to use the framework of a traditional review to conduct a dual meta-analysis *and* realist synthesis. This combined approach will offer both a quantitative and a qualitative exploration of the available evidence. Each method answers the criticism of the other; a rigorous meta-analysis will provide a numerical evaluation of intervention efficacy, while a well-conducted realist synthesis will explore study context and mechanisms. This innovative method will therefore comprehensively examine interventions that explicitly engage the family unit to increase physical activity in children. The meta-analysis may also identify specific subgroups for whom interventions are most effective, and will offer an integrated summary of available evidence. The realist synthesis will contribute to understanding factors that may mediate or moderate intervention outcomes, further informing the development of robust interventions. This may subsequently direct policymakers and practitioners towards actions that are likely to bring about positive behaviour change.

In addition to offering a more exhaustive assessment of published studies, this review will supersede existing reviews[14] [18–20] by including substantially more interventions. A brief scoping review identified a number of interventions published within the past 5 years, indicating the need for an update in this field. Exploring the most recent evidence available, particularly given the advances in physical activity measurement[28] and the application of new technologies in changing behaviour (eg, study material delivered through mobile phones or tablets),[29] will provide a contemporary summary of effective techniques. Such synthesis will then inform future work aimed at improving and maintaining physical activity levels at the critical juncture before children progress to secondary school.

### Potential limitations

The proposed work may have several limitations. As in previous reviews,[18] [20] we will be reliant on peer-reviewed published data (rather than including grey literature). While this will ensure a focused, and therefore manageable, number of studies are reviewed, it does make the review vulnerable to publication bias.[30] [31] Additionally, we have not built study author contact into our review protocol. Given the expected range of study dates of publication (ie, previous reviews have included relevant interventions published as early as 1970), we anticipate difficulties in obtaining the data. While we acknowledge the aforementioned publication bias, in reviewing only published information we avoid skewing the results in favour of those studies conducted most recently.

### Dissemination

The results of this study will be disseminated to academic and non-academic audiences through peer-reviewed publications, conferences, formal presentations to policy makers and practitioners and in formal stakeholder meetings. Evidence generated from this synthesis will also be used to inform the development of theory-driven, evidence-based interventions aimed at engaging the family to increase physical activity levels in children.

**Contributors** HEB was responsible for conducting and coordinating aspects of this study. All authors contributed considerably to editing of all written work. Article screening, general literature analyses and extraction and interpretation of data were carried out by HEB, AJA, EvS and KC. HEB and MJMC carried out the quality assessment of included studies . The development of the programme theory was done by all authors. HEB, JP and GW were involved in coding, extraction of mechanisms, and realist synthesis and interpretation of the study. HEB and EvS conducted meta-analysis of quantitative data.

**Competing interests** None.

**Provenance and peer review** Not commissioned; externally peer reviewed.

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
