## [Reviewer comments · BMJ Open]

Some articles will have been accepted based in part or entirely on reviews undertaken for other BMJ Group journals. These will be reproduced where possible.

ARTICLE DETAILS

TITLE (PROVISIONAL)	Family-based Interventions to Increase Physical Activity in Children: A Meta-Analysis and Realist Synthesis Protocol
AUTHORS	Brown, Helen Elizabeth; Atkin, Andy; Panter, Jenna; Corder, Kirsten; Wong, Geoff; Chinapaw, Mai; van Sluijs, Esther

VERSION 1 - REVIEW

REVIEWER	Tom Barnowski Professor of Pediatrics (Behavioral Nutrition & Physical Activity) USDA/ARS Children's Nutrition Research Center Department of Pediatrics Baylor College of Medicine
REVIEW RETURNED	05-May-2014

GENERAL COMMENTS	This review could quickly get out of hand. I believe the authors need to more narrowly focus the review to make it manageable. Frankly, I believe the authors should do the realist synthesis, since they appeared enamored of it, and do a separate meta-analysis to pursue the issues raised in the realist synthesis. Combining meta-analysis with a realist synthesis offers possibilities for new insights into reviews of the literature. The authors are to be applauded for their aspirations. However, combining the two forms of analysis opens a Pandora's Box of analysis possibilities. The authors need to carefully state and limit what issues they propose to address. Perhaps the existing reviews identify priority issues to address? In order to assess whether involving families in the intervention is effective, requires that studies not involving families be included in the review. It may be similarly enlightening to understand how interventions involving families are structurally different (if at all) than those not involving families. As part of the realist synthesis, it would be interesting to compare studies of similar design/structure, but one involving families and one not. The O'Connor et al. review also proposed there was moderate band of intensity of intervention that seemed optimal, within which most of the effective interventions were found. Hopefully the authors will address this interesting possibility. How was the design/structure of these moderate intensity interventions different from the more and less intense interventions? The inclusionary-exclusionary criteria are different for the meta-analysis and the realist synthesis. This is very unfortunate. Important insights in one review may come from studies not included in the
---

	other review, and thereby minimizes comparability of reviews. Why combine them if you will not be able to compare when studies contribute in each review? This is a serious flaw in the proposed research. I suggest conducting the meta-analysis first to be sure to identify those studies with the largest effects, with quantitative indications of what is contributing most to outcome, to be sure they are included in the more qualitative analyses. If additional hypotheses are generated from the realist synthesis, they can be added at a later stage to the meta-analysis. On page 6, lines 51-3, the authors state "...if relevant and of sufficient rigour, will then be extracted..." No such research quality criterion was included in the inclusionary/exclusionary criteria. The authors may be accused of "cherry-picking" studies to confirm preconceived notions if they make ad hoc inclusionary decisions. This should be avoided. It will be important to describe the studies and assess the moderation effects of family, developmental, and psychosocial/behavior change theories, using Michie's intervention procedure inventory. Journal editors have generally not encouraged detailed specifications of components of an intervention. Thus, journal articles often do not provide the details necessary for an accurate review/synthesis. The authors should build into their synthesis/review process contacting original study authors to obtain more intervention method and/or outcome details, as necessary.
--	--

REVIEWER	Melanie Henderson CHU Sainte-Justine Canada
REVIEW RETURNED	10-Jun-2014

GENERAL COMMENTS	Why were studies that targeted obese children excluded? I find the description of "studies for wider review" confusing Given that this is a protocol, questions 9 - 11 are difficult to answer, however the authors in the discussion do not address any potential limitations to their proposal - this should be added Given that the realist synthesis approach is perhaps less frequently used, I would favor enhancing the description, and being a little more explicit - eg. what is nVivo? I am not familiar with the statistical requirements of qualitative data analysis Overall, this is a well-written, interesting protocol on a clinically relevant subject matter, using novel strategies to try and capture pertinent results. I look forward to reading the results of the study!
---

VERSION 1 – AUTHOR RESPONSE

Review 1: Tom Baranowski

This review could quickly get out of hand. I believe the authors need to more narrowly focus the review to make it manageable. Frankly, I believe the authors should do the realist synthesis, since they appeared enamoured of it, and do a separate meta-analysis to pursue the issues raised in the realist synthesis. Combining meta-analysis with a realist synthesis offers possibilities for new insights into reviews of the literature. The authors are to be applauded for their aspirations. However, combining the two forms of analysis opens a Pandora's Box of analysis possibilities. The authors need to carefully state and limit what issues they propose to address. Perhaps the existing reviews identify priority issues to address?

In order to assess whether involving families in the intervention is effective, requires that studies not involving families be included in the review. It may be similarly enlightening to understand how interventions involving families are structurally different (if at all) than those not involving families. As part of the realist synthesis, it would be interesting to compare studies of similar design/structure, but one involving families and one not.

Author's response

We acknowledge the reviewer's concern regarding the potential scale of this review, particularly given the broad range of interventions that may be captured by our searches. To ensure specificity in search terms, and therefore limit extraneous results, we have set relatively definitive exclusion criteria (this has been clarified in the manuscript, see page 5, lines 27-28 for comment).

In addition, we acknowledge the challenges likely to be encountered in conducting a dual meta-analysis and realist synthesis. However, as stated in the Introduction and throughout (in particular, see page 4, lines 1-16), we believe combining the two methodologies will offer a more comprehensive examination of the literature. Since this manuscript has been submitted, additional evidence to support the potential importance of the family in increasing physical activity in children has emerged¹, further underlining the contemporary relevance of this review. Insight into the mechanisms by which interventions are effective, in addition to a traditional examination of intervention effect by meta-analysis, is essential in informing the development of future studies and overcomes the limitations of existing reviews.

The aim of the current review is not to establish whether a family component was a valuable addition to an intervention programme but to examine the effectiveness of family-based interventions per se. We agree that comparing interventions involving families with those not involving families would be insightful (and would benefit from exploration using a realist approach), but (a) this would not fulfil the review objective, and (b) may be beyond the scope of the review. Keeping focus on studies that do engage with the family will ensure a very specific and concentrated examination of an area identified as important¹⁻⁶.

The O'Connor et al. review also proposed there was moderate band of intensity of intervention that seemed optimal, within which most of the effective interventions were found. Hopefully the authors will address this interesting possibility. How was the design/structure of these moderate intensity interventions different from the more and less intense interventions?

The inclusionary-exclusionary criteria are different for the meta-analysis and the realist synthesis. This is very unfortunate. Important insights in one review may come from studies not included in the other review, and thereby minimizes comparability of reviews. Why combine them if you will not be able to compare when studies contribute in each review? This is a serious flaw in the proposed research.

We acknowledge the O'Connor review² as offering significant insight into this emerging field. As such, information regarding intervention duration, frequency and intensity will be extracted for each study, and reported using Table 2 (see manuscript). If appropriate, subgroup analyses may be conducted to explore whether an optimal 'intervention intensity' can be identified.

We describe the inclusion criteria for the general review on page 5, lines 15 – 28. Once studies have been identified by the duplicate screening process as meeting these criteria, they will be subsequently assessed for available data.

Theoretically, all studies will be able to contribute to all methods of synthesis; their contribution will only be restricted by the information they report. Only those studies providing sufficient quantitative information (i.e. mean, standard deviation and number of participants analysed) will be included in the meta-analysis (as the meta-analytic techniques proposed require these data as a minimum). Similarly, only studies that offer relevant information on context, mechanisms, or outcomes will be, by definition, included in the realist synthesis (again, techniques proposed require this information at a minimum).

The purpose of combining review techniques is to provide readers (i.e. those developing interventions to increase physical activity in children) with a comprehensive exploration of study effectiveness and underlying causal process (i.e. mechanisms). This includes both a quantitative examination of overall effect (i.e. through meta-analysis), and a more qualitative consideration of how and why programmes may work. For these data to be drawn out, a range of interventions should be included. A review comprising only studies that meet *both* of these criteria as suggested (i.e. have sufficient data for meta-analysis *and* realist synthesis) would be of limited scope (since very few interventions would be included), and not adequately reflect the breadth of interventions being conducted in this evolving field.

I suggest conducting the meta-analysis first to be sure to identify those studies with the largest effects, with quantitative indications of what is contributing most to outcome, to be sure they are included in the more qualitative analyses. If additional hypotheses are generated from the realist synthesis, they can be added at a later stage to the meta-analysis.

This is an interesting approach. However, the suggestion to conduct a meta-analysis first, to establish those studies of most use for realist synthesis, is something we disagree with.

Realist syntheses offer a narrative as to for whom an intervention might work, under what circumstances, how and why, by exploring study context, mechanisms and outcomes (and in particular, highlighting key configurations of these in individual studies).

This method poses an explanation of the outcome patterns in studies through the analysis of the data using a logic realist of enquiry. This logic of analysis deliberately focuses on trying to explain how the context within a study may have had an impact on the outcome(s) obtained. Exploring the functioning of these interventions in different contexts is important, which includes those contexts within which the intervention was not effective. By limiting the inclusion of studies to those with largest effects, valuable information may be missed (equally useful lessons may be drawn out from studies with no or limited effect).

The innovative combination method which we propose, in which neither meta-analysis nor realist synthesis precedes the other, will provide greater depth of understanding, and we believe it will therefore better inform the development of effective programmes moving forward.

On page 6, lines 51-3, the authors state "...if relevant and of sufficient rigour, will then be extracted..." No such research quality criterion was included in the inclusionary/exclusionary criteria. The authors may be accused of "cherry-picking" studies to confirm preconceived notions if they make ad hoc inclusionary decisions. This should be avoided.

The criterion here refers to the relevance of information provided in the manuscript, and does not denote study 'quality', which is assessed elsewhere. We have adjusted the sentence to increase clarity; it now reads 'if relevant and of sufficient rigour to meet RAMESES output standards' (page 7, lines 1-3).

Procedures for realist synthesis, as noted on page 6, lines 26-29, will be informed by the output of the RAMESES Project, and will be completed with methodological support from Professor Geoff Wong (author of the current realist synthesis publication standards)^{8,9}. This will include study selection, avoiding any bias in inclusion.

It will be important to describe the studies and assess the moderation effects of family, developmental, and psychosocial/behaviour change theories, using Michie's intervention procedure inventory.

We agree on the importance of describing the included studies; as such, the summary table (see Table 2 in the manuscript) will include study design employed, participants recruited, physical activity measures conducted, all elements of the intervention (and any relevant control group information), and physical activity outcomes reported. The second 'Intervention' column will, as labelled (see Table 2), include a description of any theoretical grounding identified by the study authors. We acknowledge the potential of the refined behaviour change technique taxonomy as a method of classifying studies¹⁰, and will consider this once the final study sample has been established. Additionally, moderating variables will be assessed in the meta-analysis (described on page 7, lines 20-22), and will be framed as contexts for the realist synthesis.

Journal editors have generally not encouraged detailed specifications of components of an intervention. Thus, journal articles often do not provide the details necessary for an accurate review/synthesis. The authors should build into their synthesis/review process contacting original study authors to obtain more intervention method and/or outcome details, as necessary.

We acknowledge that journal articles may not offer enough space for detailed specifications of intervention components, and have considered the suggestion of contacting study authors. However, given the expected range of study dates of publication (i.e. we are aware of relevant interventions published from 1970 through to 2013), and therefore the possible difficulty in contacting original authors, and the related risk of introducing bias, we have decided not to build this in to our review protocol. This will ensure consistency of approach, and avoid biasing the results in favour of those studies conducted most recently. If appropriate, relevant protocol papers and additional study literature will be accessed, as described on page 7, lines 6-7.

Review 2: Melanie Henderson

Author's response

Why were studies that targeted obese children excluded?

The purpose of this review is to explore interventions to increase physical activity in healthy primary-school children, with a view to understanding how we might change behaviour at a population level. Therefore, the studies included for review should each provide lessons useful in developing programmes for healthy children.

Physical activity in obese children may be influenced by different determinants (to healthy-weight children), and intervention outcomes may be driven by different mechanisms.

With this in mind, we decided to exclude all populations hypothesised to require more targeted intervention strategies (including those with a disability or pre-existing medical condition limiting their ability to be physically active).

I find the description of "studies for wider review" confusing.

This description refers to those studies which do not meet the reporting criteria for meta-analysis, and are not relevant for the realist synthesis. This phrase has been changed to "studies for general review" (see page 5, line 30 and line 31), which we believe better reflects the work being proposed.

Given that this is a protocol, questions 9 - 11 are difficult to answer, however the authors in the discussion do not address any potential limitations to their proposal - this should be added.

Given that this review has yet to be completed, it is difficult to identify limitations. We have, however, added an additional paragraph (page 8, lines 22-30), which considers some possibilities. Gaps in the literature may also limit our ability to comment on specific context-mechanism-outcome configurations, or fully explore subgroups in the meta-analysis, but we cannot comment on this prior to completion of literature screening.

Given that the realist synthesis approach is perhaps less frequently used, I would favour enhancing the description and being a little more explicit – e.g. What is nVivo?

Additional description of nVivo has been added to the text (see page 7, line 2), making clear the purpose of this software in extracting and coding qualitative data. We have also added further detail to highlight realist processes (see page 7, lines 1-13).

I am not familiar with the statistical requirements of qualitative data analysis.

Qualitative data analysis has been selected as appropriate for the realist synthesis, and will identify information in each study that refers to the context, mechanisms, and outcomes in the programme theory. Emerging themes will be highlighted, and evidence will be assessed according to contribution to the growing knowledge base (i.e. what can this study tell us about how an intervention works?).

Statistical issues common to quantitative analysis, such as sample size and statistical significance, are not relevant to qualitative data, and therefore not mentioned

Overall, this is a well-written, interesting protocol on a clinically relevant subject matter, using novel strategies to try and capture pertinent results. I look forward to reading the results of the study!

References

1. Kipping RR, Howe LD, Jago R, et al. Effect of intervention aimed at increasing physical activity, reducing sedentary behaviour, and increasing fruit and vegetable consumption in children: Active for Life Year 5 (AFLY5) school based cluster randomised controlled trial. *Bmj* 2014;**348** doi: 10.1136/bmj.g3256[published Online First: Epub Date]].
2. Activity NPHCCP. Intervention review: family and community. National Institute for Health and Clinical Excellence 2008
3. NICE. Promoting physical activity for children and young people: guidance. National Institute for Health and Clinical Excellence 2009
4. Sallis JF, Prochaska JJ, Taylor WC. A review of correlates of physical activity of children and adolescents. *Medicine and science in sports and exercise* 2000;**32**(5):963-75
5. Salmon J, Booth ML, Phongsavan P, Murphy N, Timperio A. Promoting physical activity participation among children and adolescents. *Epidemiol Rev* 2007;**29**:144-59 doi: mxm010 [pii]
10.1093/epirev/mxm010[published Online First: Epub Date]].
6. van Sluijs EM, McMinn AM, Griffin SJ. Effectiveness of interventions to promote physical activity in children and adolescents: systematic review of controlled trials. *Bmj* 2007;**335**(7622):703 doi: bmj.39320.843947.BE [pii]
10.1136/bmj.39320.843947.BE[published Online First: Epub Date]].
7. O'Connor TM, Jago R, Baranowski T. Engaging parents to increase youth physical activity a systematic review. *American journal of preventive medicine* 2009;**37**(2):141-9 doi: 10.1016/j.amepre.2009.04.020[published Online First: Epub Date]].
8. Pawson R. A realist perspective. Sage Publications 2006

9. Wong G, Greenhalgh T, Westhorp G, Buckingham J, Pawson R. RAMESES publication standards: realist syntheses. *BMC medicine* 2013;**11**:21 doi: 10.1186/1741-7015-11-21[published Online First: Epub Date]].
10. Michie S, Richardson M, Johnston M, et al. The behavior change technique taxonomy (v1) of 93 hierarchically clustered techniques: building an international consensus for the reporting of behavior change interventions. *Annals of behavioral medicine : a publication of the Society of Behavioral Medicine* 2013;**46**(1):81-95 doi: 10.1007/s12160-013-9486-6[published Online First: Epub Date]].